# EMPIRICAL LIKELIHOOD FOR FAIR CLASSIFICATION

**Pangpang Liu**
Mitchell E. Daniels, Jr. School of Business
Purdue University
West Lafayette, IN 47907, USA
`liu3364@purdue.edu`

**Yichuan Zhao**
Department of Mathematics and Statistics
Georgia State University
Atlanta, GA 30303, USA
`yichuan@gsu.edu`

## ABSTRACT

Machine learning algorithms are commonly being deployed in decision-making systems that have a direct impact on human lives. However, if these algorithms are trained solely to minimize training/test errors, they may inadvertently discriminate against individuals based on their sensitive attributes, such as gender, race or age. Recently, algorithms that ensure the fairness are developed in the machine learning community. Fairness criteria are applied by these algorithms to measure the fairness, but they often use the point estimate to assess the fairness and fail to consider the uncertainty of the sample fairness criterion once the algorithms are deployed. We suggest that assessing the fairness should take the uncertainty into account. In this paper, we use the covariance as a proxy for the fairness and develop the confidence region of the covariance vector using empirical likelihood (Owen, 1988). Our confidence region based fairness constraints for classification take uncertainty into consideration during fairness assessment. The proposed confidence region can be used to test the fairness and impose fairness constraint using the significance level as a tool to balance the accuracy and fairness. Simulation studies show that our method exactly covers the target Type I error rate and effectively balances the trade-off between accuracy and fairness. Finally, we conduct data analysis to demonstrate the effectiveness of our method.

## 1 INTRODUCTION

Machine learning methods are increasingly used in various domains to assist human decision-making. However, concerns have been raised about the fairness of these algorithmic decision-making methods. To address this issue, recent studies have proposed mechanisms to ensure the fairness of algorithmic decision systems. One such measure of fairness proposed by Zafar et al. (2017) is the covariance between sensitive attributes and the distance between the subjects' feature vectors and the decision boundary of the classifier. Zafar et al. (2019) introduced the covariance measure of decision boundary unfairness to design classifiers that are free of disparate impact and disparate mistreatment, while Nanfack et al. (2021) extended this method to decision trees. Other methods to address fairness concerns include Rényi correlation proposed by Baharlouei et al. (2020), cross-covariance operator used by Pérez-Suay et al. (2017) to measure the dependence between predictions and sensitive variables, and equalized correlations proposed by Woodworth et al. (2017) as fairness constraints. However, these methods often estimate the dependence between sensitive variables and predictions by point estimates, and do not consider the uncertainty of the estimation. Moreover, some measures of fairness are not dimensionless, making it difficult to set the threshold of the fairness constraint.

We have found solutions to these issues through the use of empirical likelihood (EL). This non-parametric method does not require assumptions about the underlying distribution and has been applied in many problems (Kallus & Uehara, 2019; Karampatziakis et al., 2020; Dai et al., 2020; Alemdjrodo & Zhao, 2022; Liu & Zhao, 2023b). In this work, instead of focusing on finding a new fairness criterion, we adopt the covariance as a tractable proxy to measure the fairness (Zafar et al., 2017), and study the problem of group fairness, which focuses on reducing the difference of favorable outcomes proportions among different sensitive groups. The covariance measure can handle both discrete and continuous sensitive attributes. Using EL, we develop a confidence region for

the fairness criterion. By using EL-based confidence region, we can consider the uncertainty of the estimation of dependence between the sensitive variable and the predictions, which is not possible with methods that rely solely on point estimates. We utilize the significance level ($\alpha \in [0, 1]$) as a threshold of the fairness constraint, which is dimensionless and provides a clear way to assess the fairness. By the confidence region, we can assess the fairness of features by checking whether $\mathbf{0}$ is included in the confidence region with Type I error rate controlled, and impose the fair constraint by constructing the domain of the parameters of interest such that $\mathbf{0}$ is included in the confidence region of the covariance vector. We do not need to set the threshold of the covariance vector, and hence the dimension of covariance has no effect on our method. In our algorithm, we balance the trade-off between the accuracy of the estimation of the parameters and the fairness constraints by the significance level. In the simulation studies, our method performs good in terms of the coverage probability and the average length of the confidence interval of the covariance. Also, our method performs better than the method in Zafar et al. (2017) in regard to the trade-off between the accuracy and fairness in some cases. In the real data analysis, we show that our method is robust to distribution shifts and can achieve simultaneous fairness by considering discrete and continuous features as protected attributes. A summary of our contributions is as follows.

(1) We propose an EL-based estimator of covariance, and establish that its limiting distribution follows the standard $\chi^2$ distribution. This conclusion does not require assumptions about the underlying distribution. Based on this limiting distribution, we can construct a confidence region for the covariance between sensitive attributes and the decision boundary, which can be used in statistical hypothesis test for group fairness.

(2) We formulate the fairness constraint using a confidence region, where the significance level $\alpha \in [0, 1]$ serves as the fairness threshold. This approach is more practical than dimensional fairness thresholds proposed in previous literature. The significance level $\alpha$, which ranges from 0 to 1 as mentioned above, can be easily determined in practice. This approach allows us to measure the fairness of both categorical and continuous features, and our algorithm can handle multiple sensitive features simultaneously to achieve the fairness. By incorporating uncertainty into the fairness measurement, our confidence region makes the fairness criterion more robust and reliable.

(3) Our work introduces a new direction for fairness constraints. We consider uncertainty when assessing and imposing fairness, which is often neglected in existing fairness criteria. Our framework can be extended to other fairness criteria, such as Rényi correlation (Baharlouei et al., 2020). We provide a discussion of applying EL to general fairness measures in Section 6.

The remainder of this paper is organized as follows. In Section 2, we review related work. In Section 3, we review the covariance as a fairness constraint, and develop the confidence region for the covariances by EL and present the EL framework for fairness. In Section 4, simulation studies are conducted to evaluate our method and compare it with other method. In Section 5, we assess our method by real data analysis. In Section 6, we conclude our work and discuss some potential extensions. The proofs and additional results are included in the appendix.

## 2 RELATED WORK

At a high level, our approach develops a confidence region for the fair criteria with the covariance as a proxy of fairness. Our framework enables auditors to assess and enforce fairness in a statistically principled manner, and can be used for statistical inference on group fairness.

Previous literature has studied statistical inference for group fairness by deriving limiting distributions of relevant statistics. However, these limiting distributions contain unknown values, rendering hypothesis tests impractical. For instance, Besse et al. (2018) showed that an estimator of disparate impact converges to a standard normal distribution that can be used to test for group disparate impact. However, the test statistic includes the unknown covariance. Similarly, Si et al. (2021) tested for group fairness using optimal transport projections and studied the asymptotic behavior of the projection distance, but the limiting distribution contains unknown values. Taskesen et al. (2021) developed a limiting distribution based on the concept of Wasserstein projection for testing group fairness, but this distribution also contains unknown values.

It is worth noting that some literature has studied inference for individual fairness. Maity et al. (2021) developed the mean of the ratio of two loss functions as a test statistic for individual fairness.

However, the test statistic is not well-defined when the classifier correctly classifies. They thus used the ratio of means (instead of the mean of the ratio) as a test statistic, which is not guaranteed to have a limiting distribution. Similarly, Xue et al. (2020) found that the asymptotic distribution for individual fairness depended on an unknown parameter.

Previous hypothesis testing methods have been problematic in practice, as the test statistics or limiting distributions have not been well-defined. Researchers have thus resorted to plugging estimates into the limiting distributions, making the tests inaccurate. Our approach, on the other hand, employs empirical likelihood ratio, which has a limiting standard $\chi^2$ distribution and does not include any unknown values. As such, it can serve as a reliable test statistic for group fairness.

## 3 STATISTICAL FRAMEWORK

In this section, we first provide a brief overview of using covariance as a fairness constraint in previous literature, and highlight its limitations. Then, we introduce our proposed method, which constructs a confidence region for the covariance vector using empirical likelihood, and uses it as a fairness constraint.

### 3.1 REVIEW OF COVARIANCE FAIRNESS CONSTRAINT

The covariance has been used as a proxy for fairness in previous literature (Zafar et al., 2017; 2019; Zink & Rose, 2020). In this section, we provide a brief overview of fair classification with covariance constraints, and highlight some of the issues that arise when working within this framework.

In binary classification problems, the goal is to find a mapping between feature vectors $\boldsymbol{x} \in \mathbb{R}^d$ and class labels $y \in \{-1, 1\}$. For margin-based classifiers, this typically involves constructing a decision boundary in the feature space that separates features based on their class labels. Let $\boldsymbol{\theta} \in \boldsymbol{\Theta}$ be the parameters of interest, and $d_{\boldsymbol{\theta}}(\boldsymbol{x}) \in \mathbb{R}$ be the distance from the feature vectors to the classifier decision boundary. Our objective is to find $\boldsymbol{\theta}$ that results in a fair classifier. The main idea is adding a constraint during the model training process. Let $\boldsymbol{s} \in \mathbb{R}^m$ be the set of sensitive attributes, such as age or gender, and $L(\boldsymbol{\theta})$ be the loss function. We denote $\boldsymbol{\sigma}_{\boldsymbol{\theta}} \in \mathbb{R}^m$ as the covariance vector between $\boldsymbol{s}$ and $d_{\boldsymbol{\theta}}(\boldsymbol{x})$, and define $\boldsymbol{\mu}_{\boldsymbol{s}} = \mathbb{E}(\boldsymbol{s})$ and $\mu_{\boldsymbol{\theta}} = \mathbb{E}(d_{\boldsymbol{\theta}}(\boldsymbol{x}))$. By the definition of covariance, we have

$$\boldsymbol{\sigma}_{\boldsymbol{\theta}} = \mathbb{E}\{(\boldsymbol{s} - \boldsymbol{\mu}_{\boldsymbol{s}})[d_{\boldsymbol{\theta}}(\boldsymbol{x}) - \mu_{\boldsymbol{\theta}}]\}.$$

Note that $\boldsymbol{\sigma}_{\boldsymbol{\theta}}$ contains $m$ elements, each of which corresponds the covariance between each sensitive feature and the classifier decision boundary. Let $\{\boldsymbol{x}_i, \boldsymbol{s}_i, y_i\}_{i=1}^n$ be independent and identically distributed samples. The sample version of $\boldsymbol{\sigma}_{\boldsymbol{\theta}}$ is

$$\hat{\boldsymbol{\sigma}}_{\boldsymbol{\theta}} = \frac{1}{n} \sum_{i=1}^n (\boldsymbol{s}_i - \bar{\boldsymbol{s}})[d_{\boldsymbol{\theta}}(\boldsymbol{x}_i) - \bar{d}_{\boldsymbol{\theta}}], \text{ with } \bar{\boldsymbol{s}} = \frac{1}{n} \sum_{i=1}^n \boldsymbol{s}_i \text{ and } \bar{d}_{\boldsymbol{\theta}} = \frac{1}{n} \sum_{i=1}^n d_{\boldsymbol{\theta}}(\boldsymbol{x}_i). \tag{1}$$

Zafar et al. (2017) estimated the parameters $\boldsymbol{\theta}$ of the decision boundary by minimizing the loss function over the training set, subject to fairness constraints. Specifically, they solved the following optimization problem,

$$\min_{\boldsymbol{\theta} \in \boldsymbol{\Theta}} L(\boldsymbol{\theta}) \text{ s.t. } \frac{1}{n} \sum_{i=1}^n (\boldsymbol{s}_i - \bar{\boldsymbol{s}}) d_{\boldsymbol{\theta}}(\boldsymbol{x}_i) \leq \boldsymbol{c}, \frac{1}{n} \sum_{i=1}^n (\boldsymbol{s}_i - \bar{\boldsymbol{s}}) d_{\boldsymbol{\theta}}(\boldsymbol{x}_i) \geq -\boldsymbol{c}, \tag{2}$$

where $\boldsymbol{c} \in \mathbb{R}^m$ is the covariance threshold. The fairness constraint in (2) uses the covariance threshold $\boldsymbol{c} \in \mathbb{R}^m$ to limit the covariance between sensitive attributes and the decision boundary, thereby balancing fairness and accuracy.

However, this approach has several limitations. Firstly, the value of $1/n \sum_{i=1}^n (\boldsymbol{s}_i - \bar{\boldsymbol{s}}) d_{\boldsymbol{\theta}}(\boldsymbol{x}_i)$ depends on the samples, while $\boldsymbol{c}$ is fixed. Consequently, this method fails to account for the uncertainty in the covariance estimation in the fairness constraint. Secondly, since the covariance is not scale-free and $\boldsymbol{c}$ is a $m$ dimensional vector, determining the threshold $\boldsymbol{c}$ that balances accuracy and fairness becomes difficult. Thirdly, when $\boldsymbol{c} = 0$ and $d_{\boldsymbol{\theta}}(\boldsymbol{x}) = \boldsymbol{\theta}^\top \boldsymbol{x}$, the constraint reduces to a system of $m$ equations and $d$ parameters, which can possibly only have solutions $\boldsymbol{\theta} = \boldsymbol{0}$ if $m \geq d$. Lastly, the confidence region of the covariance vector is not constructed, and statistical inference is not addressed in this existing framework. To address these limitations, we propose a confidence region-based fairness constraint for classification using empirical likelihood.

### 3.2 EMPIRICAL LIKELIHOOD FOR COVARIANCE

In this section, we develop the confidence region for the covariance vector $\boldsymbol{\sigma_\theta}$ by empirical likelihood. Empirical likelihood is used to construct generalized likelihood ratio test statistics and corresponding confidence regions without specifying parametric models for the data. Comprehensive reviews about empirical likelihood can be found in Owen (2001) and Liu & Zhao (2023a).

We apply the EL method based on influence functions, which avoids the involvement of nuisance parameters and achieve a limiting distribution without any unknown quantities. For a fixed $\boldsymbol{\theta}$, by Equation (1), simple algebra yields that

$$\sqrt{n}(\hat{\boldsymbol{\sigma}}_{\boldsymbol{\theta}} - \boldsymbol{\sigma}_{\boldsymbol{\theta}}) = \sqrt{n}\left\{\frac{1}{n}\sum_{i=1}^{n}(\boldsymbol{s}_i - \bar{\boldsymbol{s}})[d_{\boldsymbol{\theta}}(\boldsymbol{x}_i) - \bar{d}_{\boldsymbol{\theta}}] - \boldsymbol{\sigma}_{\boldsymbol{\theta}}\right\}$$

$$= \frac{1}{\sqrt{n}}\sum_{i=1}^{n}\{(\boldsymbol{s}_i - \boldsymbol{\mu_s})[d_{\boldsymbol{\theta}}(\boldsymbol{x}_i) - \mu_{\boldsymbol{\theta}}] - \boldsymbol{\sigma}_{\boldsymbol{\theta}}\} - \sqrt{n}(\bar{\boldsymbol{s}} - \boldsymbol{\mu_s})(\bar{d}_{\boldsymbol{\theta}} - \mu_{\boldsymbol{\theta}}) \quad (3)$$

$$= \frac{1}{\sqrt{n}}\sum_{i=1}^{n}\{(\boldsymbol{s}_i - \boldsymbol{\mu_s})[d_{\boldsymbol{\theta}}(\boldsymbol{x}_i) - \mu_{\boldsymbol{\theta}}] - \boldsymbol{\sigma}_{\boldsymbol{\theta}}\} + o_p(1).$$

In the last equation, the second term is $o_p(1)$ because $\sqrt{n}(\bar{\boldsymbol{s}} - \boldsymbol{\mu_s}) = O_p(1)$ and $\sqrt{n}(\bar{d}_{\boldsymbol{\theta}} - \mu_{\boldsymbol{\theta}}) = O_p(1)$ by the central limit theorem. Equation (3) implies that the $i$-th influence function of $\hat{\boldsymbol{\sigma}}_{\boldsymbol{\theta}}$ is $(\boldsymbol{s}_i - \boldsymbol{\mu_s})[d_{\boldsymbol{\theta}}(\boldsymbol{x}_i) - \mu_{\boldsymbol{\theta}}] - \boldsymbol{\sigma}_{\boldsymbol{\theta}}$. Motivated by Zheng et al. (2012), we construct the empirical likelihood ratio at $\boldsymbol{\sigma}_{\boldsymbol{\theta}}$ based on the estimated influence function as follows,

$$R(\boldsymbol{\sigma}_{\boldsymbol{\theta}}) = \sup_{p_1,\ldots,p_n}\left\{\Pi_{i=1}^{n}(np_i)\bigg|\sum_{i=1}^{n}p_i = 1, \sum_{i=1}^{n}p_i(\boldsymbol{s}_i - \bar{\boldsymbol{s}})[d_{\boldsymbol{\theta}}(\boldsymbol{x}_i) - \bar{d}_{\boldsymbol{\theta}}] = \boldsymbol{\sigma}_{\boldsymbol{\theta}}, p_i \geq 0\right\}, \quad (4)$$

where $p_i$ is the probability placed on the $i$-th sample. Using the Lagrange multiplier method to solve (4), we obtain

$$p_i = \frac{1}{n}\frac{1}{1 + \boldsymbol{\lambda}^\top\{(\boldsymbol{s}_i - \bar{\boldsymbol{s}})[d_{\boldsymbol{\theta}}(\boldsymbol{x}_i) - \bar{d}_{\boldsymbol{\theta}}] - \boldsymbol{\sigma}_{\boldsymbol{\theta}}\}},$$

where $\boldsymbol{\lambda}$ is the Lagrange multiplier. Hence, the empirical log-likelihood ratio can be obtained as follows,

$$-2\log R(\boldsymbol{\sigma}_{\boldsymbol{\theta}}) = 2\sum_{i=1}^{n}\log\{1 + \boldsymbol{\lambda}^\top[(\boldsymbol{s}_i - \bar{\boldsymbol{s}})[d_{\boldsymbol{\theta}}(\boldsymbol{x}_i) - \bar{d}_{\boldsymbol{\theta}}] - \boldsymbol{\sigma}_{\boldsymbol{\theta}}]\}, \quad (5)$$

where $\boldsymbol{\lambda}$ satisfies

$$\frac{1}{n}\sum_{i=1}^{n}\frac{(\boldsymbol{s}_i - \bar{\boldsymbol{s}})[d_{\boldsymbol{\theta}}(\boldsymbol{x}_i) - \bar{d}_{\boldsymbol{\theta}}] - \boldsymbol{\sigma}_{\boldsymbol{\theta}}}{1 + \boldsymbol{\lambda}^\top\{(\boldsymbol{s}_i - \bar{\boldsymbol{s}})[d_{\boldsymbol{\theta}}(\boldsymbol{x}_i) - \bar{d}_{\boldsymbol{\theta}}] - \boldsymbol{\sigma}_{\boldsymbol{\theta}}\}} = \boldsymbol{0}. \quad (6)$$

Now, we establish the Wilk's theorem for the empirical likelihood ratio as follows (Zheng et al., 2012).

**Theorem 3.1** *Under conditions C1-C4 in Appendix A, for a fixed $\boldsymbol{\theta}$, at the corresponding true value $\boldsymbol{\sigma}_{\boldsymbol{\theta}}$, we have*

$$-2\log R(\boldsymbol{\sigma}_{\boldsymbol{\theta}}) \xrightarrow{D} \chi_m^2,$$

*where $\chi_m^2$ is the chi-squared distribution with $m$ degrees of freedom, and $\xrightarrow{D}$ denotes the convergence in distribution.*

Theorem 3.1 provides the asymptotic property of the log-empirical likelihood ratio. The limiting distribution of $-2\log R(\boldsymbol{\sigma}_{\boldsymbol{\theta}})$ is a standard $\chi^2$ distribution without any unknown quantities. In contrast, previous fairness literature (Besse et al., 2018; Xue et al., 2020; Si et al., 2021) derived the limiting distributions that contain unknown quantities, which can reduce the accuracy of statistical inference when the estimates of the unknown parameters are used.

We derive the confidence region of $\boldsymbol{\sigma}_{\boldsymbol{\theta}}$ based on Theorem 3.1 for a fixed $\boldsymbol{\theta}$. The EL confidence region with $(1 - \alpha)$ confidence level for $\boldsymbol{\sigma}_{\boldsymbol{\theta}}$ at a given $\boldsymbol{\theta}$ is constructed as

$$I_{EL}(\boldsymbol{\sigma}_{\boldsymbol{\theta}}) = \{\tilde{\boldsymbol{\sigma}} \in \mathbb{R}^m : -2\log R(\tilde{\boldsymbol{\sigma}}) \leq \chi_m^2(\alpha)\}, \quad (7)$$

where $\chi_m^2(\alpha)$ is the upper $\alpha$-quantile of $\chi_m^2$. Equation (7) can be used to impose a fairness constraint during the model training process and forms the basis for the fairness inference.

### 3.3 FAIRNESS VIA EMPIRICAL LIKELIHOOD

In this section, we develop a novel framework to address the limitations of previous methods and achieve the fairness in classification. We formulate the EL-based fairness framework. Our framework can be used for statistical inference and as well for imposing fairness constraint during the model training process.

To conduct statistical inference for fairness, we utilize the confidence region defined in Equation (7). We test for fairness by checking whether $\mathbf{0}$ is included in $I_{EL}(\boldsymbol{\sigma_\theta})$ for statistical inference. If $\mathbf{0} \in I_{EL}(\boldsymbol{\sigma_\theta})$, i.e., $-2 \log R(\mathbf{0}) \leq \chi_m^2(\alpha)$, it means that $\mathbf{0}$ is in the $(1 - \alpha)$ confidence region of $\boldsymbol{\sigma}_\theta$, and correspondingly, the linear dependence between the sensitive features and the decision boundary is not significant at the $\alpha$ significance level.

To obtain a fair classifier, we can impose the fairness constraint during the training process by finding the decision boundary parameter $\boldsymbol{\theta}$ such that $\mathbf{0}$ is included in $I_{EL}(\boldsymbol{\sigma_\theta})$. Our approach differs from previous literature as we reduce the uncertainty associated with the point estimate by incorporating the confidence region into the training process. To find the optimal decision boundary parameters that satisfy the fairness constraint, we minimize the corresponding loss function $L(\boldsymbol{\theta})$ over the training set under the fairness constraints given by the following optimization problem,

$$\min_{\boldsymbol{\theta} \in \boldsymbol{\Theta}, \boldsymbol{\lambda} \in \mathbb{R}^m} L(\boldsymbol{\theta}) \text{ s.t. } \quad 2 \sum_{i=1}^n \log\{1 + \boldsymbol{\lambda}^\top (\boldsymbol{s}_i - \bar{\boldsymbol{s}})[d_{\boldsymbol{\theta}}(\boldsymbol{x}_i) - \bar{d}_{\boldsymbol{\theta}}]\} \leq \chi_m^2(\alpha),$$

$$\frac{1}{n} \sum_{i=1}^n \frac{(\boldsymbol{s}_i - \bar{\boldsymbol{s}})[d_{\boldsymbol{\theta}}(\boldsymbol{x}_i) - \bar{d}_{\boldsymbol{\theta}}]}{1 + \boldsymbol{\lambda}^\top (\boldsymbol{s}_i - \bar{\boldsymbol{s}})[d_{\boldsymbol{\theta}}(\boldsymbol{x}_i) - \bar{d}_{\boldsymbol{\theta}}]} = \mathbf{0}. \tag{8}$$

The constraint in (8) is obtained by plugging $\boldsymbol{\sigma_\theta} = \mathbf{0}$ into the equations (5) and (6). Intuitively, as the value of $\alpha$ decreases, the corresponding $\chi_m^2(\alpha)$ increases, resulting in a looser fairness constraint and higher accuracy of parameter estimation. Conversely, a larger $\alpha$ value leads to a stricter fairness constraint, but lower accuracy of parameter estimation. In the extreme case $\alpha = 0$, $I_{EL}(\boldsymbol{\sigma})$ becomes the entire space $\mathbb{R}^m$, which means there is no fairness constraint. Thus, the significance level $\alpha$ can serve as a trade-off indicator between accuracy and fairness, and unconstrained model can be viewed as a special case of our method. Note that the constraint in (2) is symmetric, while the constraint in our framework is asymmetric, which contains more information. Our framework uses the covariance as a fairness proxy, and it can be easily extended to other fair criteria by developing a confidence region for those criteria using EL. We present the use of our approach with the binary logistic regression in Appendix B as an example.

## 4 SIMULATION

In this section, we evaluate our method by simulation studies. We first examine the limiting distribution provided in Theorem 3.1 empirically by the coverage probability and the confidence interval. Then, we explore the trade-off between the accuracy and fairness using our method and compare it with the method in Zafar et al. (2017) by experiments.

### 4.1 COVERAGE PROBABILITY AND CONFIDENCE INTERVAL

In this section, we examine the limiting distribution from Theorem 3.1 in terms of the coverage probability and the confidence interval. We consider a linear decision boundary $d_{\boldsymbol{\theta}}(\boldsymbol{x}) = \theta_1 x_1 + \theta_2 x_2$, with $\boldsymbol{\theta} = (\theta_1, \theta_2)^\top$. One sensitive feature $\boldsymbol{s} \in \mathbb{R}$ is continuous and follows a Gaussian distribution $N(0, 2)$, while $x_1$ and $x_2$ are both from the Gaussian distribution $N(0, 1)$. The covariance between $\boldsymbol{s}$ and $x_1$ is $1/2$, the covariance between $\boldsymbol{s}$ and $x_2$ is $1$, and the covariance between $x_1$ and $x_2$ is $0$. The sample size $n$ varies from 100 to 1200, and we evaluate the confidence interval for the covariance $\boldsymbol{\sigma_\theta}$ between $\boldsymbol{s}$ and $d_{\boldsymbol{\theta}}(\boldsymbol{x})$ using EL at $\alpha = 0.05$, under four scenarios: $\boldsymbol{\theta} = (1/4, 1/3)^\top$ with $\boldsymbol{\sigma_\theta} = 11/24$, $\boldsymbol{\theta} = (1/2, 2/3)^\top$ with $\boldsymbol{\sigma_\theta} = 11/12$, $\boldsymbol{\theta} = (2, 1)^\top$ with $\boldsymbol{\sigma_\theta} = 2$, and $\boldsymbol{\theta} = (3, 2)^\top$ with $\boldsymbol{\sigma_\theta} = 7/2$. We repeat each experiment 2000 times, and present the coverage probability (CP), the average lower bound (LB), the average upper bound (UB) and the average length (AL) of confidence interval for $\boldsymbol{\sigma_\theta}$ in Table 1. The results for the scenarios, $\boldsymbol{\theta} = (2, 1)^\top$ with $\boldsymbol{\sigma_\theta} = 2$, and $\boldsymbol{\theta} = (3, 2)^\top$ with $\boldsymbol{\sigma_\theta} = 7/2$ are shown in Appendix C.

Table 1: Coverage probability and confidence interval

| | $\boldsymbol{\theta} = (1/4, 1/3)^\top$ | | | | $\boldsymbol{\theta} = (1/2, 2/3)^\top$ | | | |
|---|---|---|---|---|---|---|---|---|
| $n$ | CP | LB | UB | AL | CP | LB | UB | AL |
| 100 | 0.935 | 0.330 | 0.621 | 0.291 | 0.935 | 0.661 | 1.243 | 0.582 |
| 200 | 0.941 | 0.363 | 0.570 | 0.207 | 0.941 | 0.726 | 1.139 | 0.413 |
| 500 | 0.943 | 0.396 | 0.527 | 0.131 | 0.943 | 0.791 | 1.053 | 0.262 |
| 800 | 0.947 | 0.408 | 0.512 | 0.104 | 0.947 | 0.816 | 1.023 | 0.207 |
| 1200 | 0.944 | 0.417 | 0.501 | 0.085 | 0.944 | 0.834 | 1.003 | 0.169 |

The results in Table 1 indicate that the coverage probabilities under different scenarios are close to 0.95. Under different scenarios, the true values of $\boldsymbol{\sigma_\theta}$ are between the lower bound and the upper bound at different sample size $n$, and the average length for $\boldsymbol{\sigma_\theta}$ decreases as the sample size $n$ increases. In summary, our EL-based method provides reliable confidence intervals for the covariance.

## 4.2 TRADE-OFF BETWEEN ACCURACY AND FAIRNESS

In this section, we investigate the trade-off between accuracy and fairness, and compare our approach with the method in Zafar et al. (2017). To facilitate the comparison, we employ the settings proposed by Zafar et al. (2017). We generate 2000 binary class labels uniformly at random and assign a 2-dimensional feature vector to each label by drawing samples from two distinct Gaussian distributions: $p(\boldsymbol{x}|y = 1) = N([2; 2], [5, 1; 1, 5])$ and $p(\boldsymbol{x}|y = -1) = N([-2; -2], [10, 1; 1, 3])$. We use $\boldsymbol{x}' = [\cos(\phi), -\sin(\phi); \sin(\phi), \cos(\phi)]\boldsymbol{x}$ as a rotation of the feature vector $\boldsymbol{x}$, and draw the one-dimensional sensitive attribute $\boldsymbol{s}$ from a Bernoulli distribution, $p(\boldsymbol{s} = 1) = p(\boldsymbol{x}'|y = 1)/[p(\boldsymbol{x}'|y = 1) + p(\boldsymbol{x}'|y = -1)]$. The value of $\phi$ controls the correlation between the sensitive attribute and the class labels. We choose $\phi = \pi/3$, $\alpha = 0.05$.

Table 2: Trade-off between model performance and fairness

| | Zafar et al. (2017) | | | | | EL-based fairness | | | | | | |
|---|---|---|---|---|---|---|---|---|---|---|---|---|
| $\boldsymbol{c}/\alpha$ | ACC | F1 | $p\%$ | DP | EO | ACC | F1 | $p\%$ | DP | EO | LB | UB |
| 0 | 0.825 | 0.827 | 92.308 | -0.029 | 0.083 | 0.847 | 0.845 | 68.959 | 0.188 | 0.366 | 0.199 | 0.474 |
| 0.1 | 0.842 | 0.840 | 78.548 | 0.094 | 0.240 | 0.843 | 0.843 | 81.346 | 0.065 | 0.207 | -0.069 | 0.188 |
| 0.2 | 0.840 | 0.838 | 73.734 | 0.124 | 0.299 | 0.842 | 0.842 | 83.077 | 0.050 | 0.188 | -0.084 | 0.171 |
| 0.3 | 0.843 | 0.841 | 68.821 | 0.190 | 0.366 | 0.840 | 0.841 | 83.654 | 0.050 | 0.182 | -0.093 | 0.159 |
| 0.4 | 0.847 | 0.845 | 68.959 | 0.188 | 0.366 | 0.833 | 0.834 | 87.016 | 0.025 | 0.142 | -0.101 | 0.150 |
| 0.5 | 0.847 | 0.845 | 68.959 | 0.188 | 0.366 | 0.835 | 0.836 | 87.604 | 0.011 | 0.135 | -0.107 | 0.142 |
| 0.6 | 0.847 | 0.845 | 68.959 | 0.188 | 0.366 | 0.833 | 0.835 | 89.368 | -0.004 | 0.116 | -0.113 | 0.135 |
| 0.7 | 0.847 | 0.845 | 68.959 | 0.188 | 0.366 | 0.832 | 0.833 | 89.941 | -0.016 | 0.109 | -0.118 | 0.129 |
| 0.8 | 0.847 | 0.845 | 68.959 | 0.188 | 0.366 | 0.830 | 0.832 | 90.533 | -0.016 | 0.103 | -0.123 | 0.123 |
| 0.9 | 0.847 | 0.845 | 68.959 | 0.188 | 0.366 | 0.827 | 0.829 | 91.716 | -0.016 | 0.090 | -0.127 | 0.117 |

We partition the data into a training set (70%) and a test set (30%) and fit a logistic model (Appendix B). We use accuracy (ACC), F1-score (F1) as the performance metrics of the model, $p\%$-rule[1] ($p\%$), demographic disparity (DP) and equal opportunity (EO) as the fairness metrics. Table 2 displays the relationship between the model performance and fairness at various fairness thresholds using the method in Zafar et al. (2017) and our method EL-based fairness. The first column in Table 2 is the parameter $\boldsymbol{c}$ for Zafar et al. (2017) and the parameter $\alpha$ for our method. The results are calculated from the test data, and for our method, we also calculate the lower bound (LB) and the upper bound (UB) of the covariance. We draw the following conclusions from Table 2:

(1) For the method in Zafar et al. (2017), the threshold $\boldsymbol{c}$ is defined in (2). As $\boldsymbol{c}$ increases, the model becomes more accurate but less fair. As $\boldsymbol{c}$ surpasses 0.4, accuracy and fairness remain stable,

---

[1]According to the $p\%$-rule (Biddle, 2005), the ratio between the proportion of subjects with a certain sensitive attribute value assigned the positive decision outcome and the proportion of subjects without that value who receive the positive outcome should not be less than $p/100$. The 80%-rule is generally used in practice

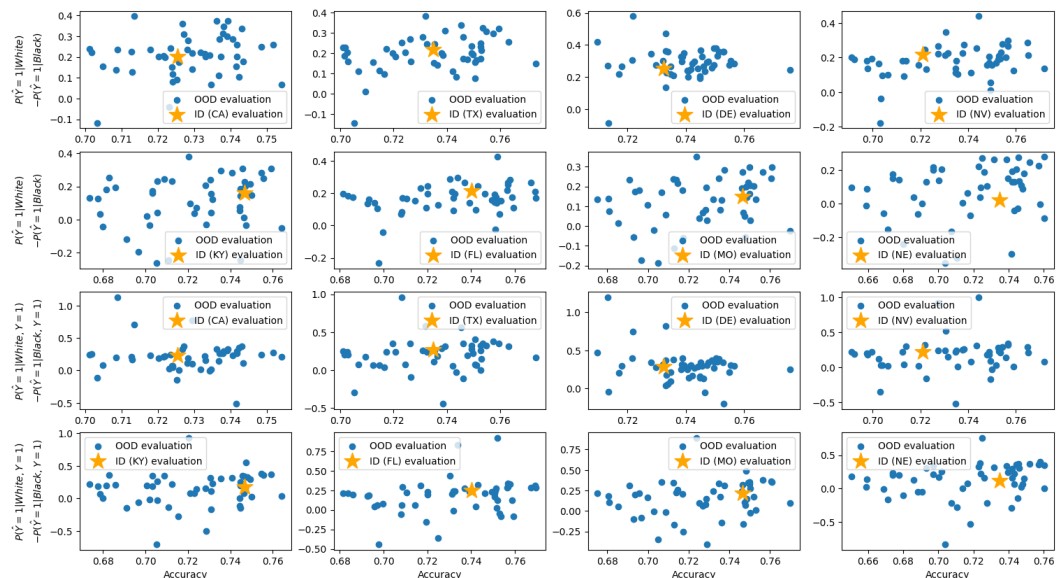

Figure 1: Training on one state (ID (state) evaluation) and testing on other 49 states (OOD evaluation) for the income task. The $x$-axis represents the accuracy. Top 2 lines: The $y$-axis represents the demographic disparity. Bottom 2 lines: The $y$-axis represents the equal opportunity.

indicating that 0.4 is an upper bound for $c$ as a trade-off indicator in this scenario. As $c$ can take any non-negative value, finding a suitable threshold in practice is challenging.

(2) For our approach, we use the significance level $\alpha$ to balance the accuracy and fairness. The value of $\alpha$ varies from 0 to 1, which is easy to control. When $\alpha = 0$, the model is unconstrained, and 0 is not between the lower bound and the upper bound. As $\alpha$ increases, the model becomes more fair but generally less accurate. The results show that $\alpha$ balances the accuracy and fairness effectively. Our method provides the confidence interval, which can be used in statistical inference.

(3) Because of the existence of the trade-off between the performance and fairness, a model is deemed superior if it achieves a better performance and a higher fairness than others. Our method achieves better model performance and higher fairness than the method in Zafar et al. (2017) in the following cases: $\alpha = 0.1$ vs. $c = 0.3$, $\alpha = 0.2$ vs. $c = 0.1$, and $\alpha = 0.3$ vs. $c = 0.2$, $\alpha = 0.1$ vs. $c = 0.1$, $\alpha = 0.1$ vs. $c = 0.2$, and $\alpha = 0.2$ vs. $c = 0.2$. Table 2 reveals no cases in which the method in Zafar et al. (2017) performs better.

## 5 DATA ANALYSIS

In this section, we conduct experiments to assess the efficacy of our method using real data. Firstly, we apply our method on the ACS PUMS datasets (Ding et al., 2021), which encompass distribution shifts, and we find our method is robust to distribution shifts. Secondly, we consider multiple sensitive attributes of the German credit dataset (Dua & Graff, 2019) and show that our method achieves simultaneous fairness.

### 5.1 ROBUST TO DISTRIBUTION SHIFTS

We leverage the ACS PUMS datasets, which explicitly encompass distribution shifts, where the distribution of the test dataset differs from the training dataset. We study the performance of our method under the geographic distribution shift, where we train a model on one state and test it on another as Ding et al. (2021) did. This dataset is accessible through the "folktables" Python package. In our experiments, we construct a logistic model (Appendix B) for the income prediction task: predicting income above 50000. The dataset encompasses 10 features along with a single label denoting income above 50000. The size of the training dataset varies from 4713 to 195665.

We designate race as the sensitive attribute. In Table 3 and Figure 1, we present the accuracy (ACC), demographic disparity (DP), and equal opportunity (EO) metrics with respect to race, for the income prediction. Each figure includes two crucial facets: the in-distribution (ID) results, signifying the training outcomes, and the out-of-distribution (OOD) results, indicating the performance across 49 other states during testing. The insights derived from these experiments are as follows.

(1) Figure 3 in Ding et al. (2021) underscores a marked separation between the in-distribution and out-of-distribution results. This divergence signifies a severe violation in model accuracy and demographic parity. In contrast, our results in Figure 1, depict OOD outcomes distributed around the ID results in a random pattern under the demographic parity and equal opportunity metrics. This pronounced alignment strongly signifies the robustness of our method to distribution shifts.

(2) Table 3 presents the evaluations for $\alpha = 0.05$ and $\alpha = 0.5$ using our method. The ID outcomes denote training results, while the OOD represents the averaged evaluations across the 49 states during testing. A comparison between the ID and OOD results reveals small fluctuations, thereby underlining the robustness of our approach under distribution shifts. Intriguingly, the variation between $\alpha = 0.05$ and $\alpha = 0.5$ elucidates a consistent trend: higher fairness at the cost of marginally reduced accuracy. This consistent pattern reaffirms the efficacy of the underlying balancing role of $\alpha$. Table 4 shows the results without fairness constraint. Upon comparing the results presented in Table 3 and Table 4, we find our method leads to a decrease in both DP and EO in most cases at a minor cost to accuracy.

Table 3: Results using our method

|  | $\alpha = 0.05$ | | | | | | $\alpha = 0.5$ | | | | | |
|  | ID (training) | | | OOD (test) | | | ID (training) | | | OOD (test) | | |
| State | ACC | DP | EO | ACC | DP | EO | ACC | DP | EO | ACC | DP | EO |
|---|---|---|---|---|---|---|---|---|---|---|---|---|
| CA | 0.7254 | 0.2008 | 0.2309 | 0.7287 | 0.2073 | 0.2120 | 0.7243 | 0.1991 | 0.2279 | 0.7284 | 0.2034 | 0.2123 |
| TX | 0.7348 | 0.2195 | 0.2637 | 0.7346 | 0.1982 | 0.2085 | 0.7335 | 0.2195 | 0.2621 | 0.7340 | 0.1905 | 0.1988 |
| DE | 0.7324 | 0.2515 | 0.2865 | 0.7396 | 0.2895 | 0.2930 | 0.7286 | 0.2176 | 0.2780 | 0.7383 | 0.2665 | 0.2619 |
| NV | 0.7209 | 0.2193 | 0.2230 | 0.7334 | 0.1781 | 0.1796 | 0.7167 | 0.1944 | 0.2076 | 0.7319 | 0.1657 | 0.1542 |
| KY | 0.7468 | 0.1591 | 0.1802 | 0.7225 | 0.1046 | 0.1405 | 0.7456 | 0.1444 | 0.1523 | 0.7194 | 0.0831 | 0.1287 |
| FL | 0.7403 | 0.2157 | 0.2517 | 0.7315 | 0.1655 | 0.1789 | 0.7383 | 0.2115 | 0.2454 | 0.7301 | 0.1587 | 0.1743 |
| MO | 0.7466 | 0.1479 | 0.2178 | 0.7260 | 0.1242 | 0.1532 | 0.7460 | 0.1435 | 0.2063 | 0.7233 | 0.1024 | 0.1407 |
| NE | 0.7346 | 0.0224 | 0.1153 | 0.7168 | 0.0713 | 0.1206 | 0.7311 | -0.0137 | 0.0417 | 0.7128 | 0.0436 | 0.1096 |

## 5.2 MULTIPLE SENSITIVE ATTRIBUTES

In this section, we evaluate the performance of our proposed method on a real-world dataset, considering two sensitive features simultaneously. Specifically, we use the German credit dataset (Dua & Graff, 2019), which contains 1000 instances of bank account holders and is commonly used for risk assessment prediction. Each sample is described by 13 categorical, 7 numerical, and 1 binary attribute. Two sensitive attributes, gender and age, are considered in the evaluation. While previous studies have binarized the age feature by thresholding at 25 as in Kamiran & Calders (2009), we

Table 4: Results without constraint

|  | ID (training) | | | OOD (test) | | |
| State | ACC | DP | EO | ACC | DP | EO |
|---|---|---|---|---|---|---|
| CA | 0.7559 | 0.1996 | 0.2641 | 0.7296 | 0.3867 | 0.3156 |
| TX | 0.7538 | 0.2529 | 0.3355 | 0.7424 | 0.3374 | 0.3749 |
| DE | 0.7382 | 0.2760 | 0.3334 | 0.7419 | 0.3410 | 0.3540 |
| NV | 0.7477 | 0.2798 | 0.3011 | 0.7400 | 0.3547 | 0.4019 |
| KY | 0.7540 | 0.2375 | 0.3306 | 0.7388 | 0.2732 | 0.3408 |
| FL | 0.7565 | 0.2982 | 0.3698 | 0.7419 | 0.2944 | 0.3287 |
| MO | 0.7511 | 0.1915 | 0.2664 | 0.7375 | 0.2582 | 0.3321 |
| NE | 0.7439 | 0.2386 | 0.4966 | 0.7345 | 0.2542 | 0.3507 |

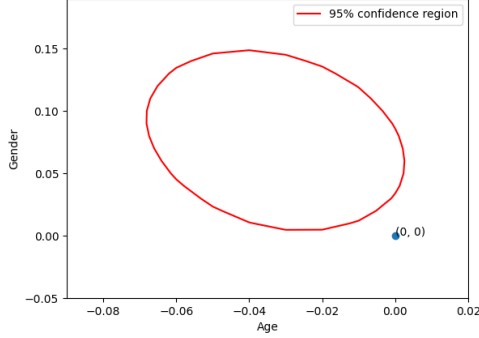 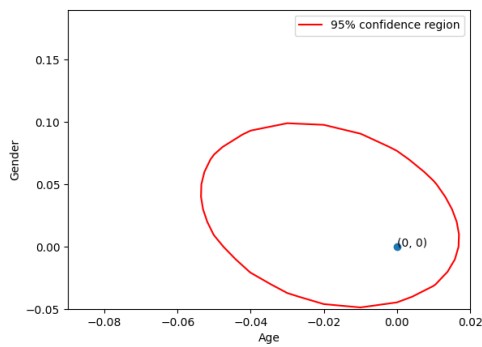

(a) Covariance contour under unconstrained model      (b) Covariance contour under constrained model

Figure 2: The $95\%$ confidence region for constrained model and unconstrained model. The $x$-axis is the covariance between age and the decision boundary. The $y$-axis is the covariance between gender and the decision boundary. The red line represent covariance contour at $95\%$ confidence level.

avoid this step to preserve the information contained in the continuous age feature. To avoid the effect of sample size on the confidence region, we split the dataset into equal training and test sets.

We train a logistic model (Appendix B) on the dataset, using both gender (binary) and age (continuous) as protected attributes. Without considering fairness constraints, the model achieves an accuracy of $0.742$. When we apply our EL-based fairness constraint, the accuracy decreases slightly to $0.732$. Figure 2 shows the $95\%$ confidence region under the constrained and unconstrained models, respectively. We observe that, without the fairness constraint, the covariance vector $(0,0)^\top$ falls outside the $95\%$ confidence region, while with the fairness constraint, the covariance vector $(0,0)^\top$ falls within the $95\%$ confidence region. This suggests that our method effectively achieves a higher level of fairness while maintaining a reasonable level of accuracy.

## 6   DISCUSSION

In this paper, we propose a confidence region for the covariance vector between the sensitive features and classifier decision boundary by empirical likelihood. Our method can be used in statistical inference for the group fairness, and imposing a fairness constraint in the process of training model. We can also develop the confidence region of the covariance vector by jackknife empirical likelihood (Jing et al., 2009). Recall that $\hat{\boldsymbol{\sigma}}_{\boldsymbol{\theta}} = 1/n \sum_{i=1}^{n} (\boldsymbol{s}_i - \bar{\boldsymbol{s}})[d_{\boldsymbol{\theta}}(\boldsymbol{x}_i) - \bar{d}_{\boldsymbol{\theta}}]$. We define the jackknife pseudo values $\hat{\boldsymbol{\sigma}}_j = n\hat{\boldsymbol{\sigma}}_{\boldsymbol{\theta}} - (n-1)\hat{\boldsymbol{\sigma}}_{\boldsymbol{\theta}}^{(-j)}$ of $\boldsymbol{\sigma}_{\boldsymbol{\theta}}$ for $j = 1, \cdots, n$, where $\hat{\boldsymbol{\sigma}}_{\boldsymbol{\theta}}^{(-j)}$ is calculated by removing the $j$-th element. Then the jackknife EL ratio at $\boldsymbol{\sigma}_{\boldsymbol{\theta}}$ is defined as

$$R^J(\boldsymbol{\sigma}_{\boldsymbol{\theta}}) = \max \left[ \Pi_{j=1}^n (np_j) : \sum_{j=1}^{n} p_j \hat{\boldsymbol{\sigma}}_j = \boldsymbol{\sigma}_{\boldsymbol{\theta}}, \sum_{j=1}^{n} p_j = 1, p_j \geq 0 \right],$$

where $p_j$ is the probability placed on the $j$-the element. Notably, the logarithm of the jackknife EL ratio, denoted as $-2\log R^J(\boldsymbol{\sigma}_{\boldsymbol{\theta}})$, may be proved to converge to the $\chi_d^2$ distribution. In this paper, we use the covariance as a fairness criterion. However, covariance is not a perfect fairness criterion. We can apply EL to other non-linear fairness criteria, such as the cross-covariance operator (Pérez-Suay et al., 2017), equalized correlations (Woodworth et al., 2017), the Rényi correlation (Baharlouei et al., 2020) and the Hirschfeld-Gebelein-Rényi maximum correlation coefficient (Mary et al., 2019). Some valuable references of adapting our framework to non-linear measures are Qin & Lawless (1994); Kremer et al. (2022); Nabi et al. (2022). If the empirical estimating equation contains non-$i.i.d.$ elements, we can develop the confidence region of the fairness measures by using jackknife empirical likelihood (Jing et al., 2009; Zhao et al., 2015; Li et al., 2016; Sang et al., 2020; Matsushita & Otsu, 2020; Liu & Zhao, 2023a). We can also tackle fairness concerns by incorporating the prior information using Bayesian empirical likelihood (Lazar, 2003; Chaudhuri et al., 2017; Zhao et al., 2020) and Bayesian jackknife empirical likelihood (Cheng & Zhao, 2019).

## ACKNOWLEDGMENTS

Yichuan Zhao acknowledges the support from Simons Foundation (Grant Number 638679), and the National Science Foundation (Grant Number 2317533).

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

# A PROOF OF THEOREM 3.1

For ease of expressing the conditions needed to derive the limiting distribution of the empirical log-likelihood ratio, we denote the true parameters as

$$\boldsymbol{v_\theta} = (\boldsymbol{\sigma_\theta^\top}, \boldsymbol{\mu_s^\top}, \mu_\theta)^\top \in \mathbb{R}^{2m+1},$$

the correspond generic values as

$$\boldsymbol{v} = (\boldsymbol{\sigma^\top}, \boldsymbol{\mu_1^\top}, \mu_2)^\top \in \mathbb{R}^{2m+1},$$

and $2m + 1$ functions as

$$H_{\boldsymbol{\theta}}(\boldsymbol{s}, \boldsymbol{x}; \boldsymbol{v}) = \begin{pmatrix} sd_{\boldsymbol{\theta}}(\boldsymbol{x}) - \boldsymbol{\mu_1}\mu_2 - \boldsymbol{\sigma} \\ \boldsymbol{s} - \boldsymbol{\mu_1} \\ d_{\boldsymbol{\theta}}(\boldsymbol{x}) - \mu_2 \end{pmatrix}.$$

To establish theoretical results, we assume that the following conditions hold.

C1. $\mathbb{E}H_{\boldsymbol{\theta}}(\boldsymbol{s}, \boldsymbol{x}; \boldsymbol{v}) \neq \boldsymbol{0}$ for $\boldsymbol{v} \neq \boldsymbol{v_\theta}$.

C2. $H_{\boldsymbol{\theta}}(\boldsymbol{s}, \boldsymbol{x}; \boldsymbol{v})$ has continuous first-order derivative with respect to $\boldsymbol{v}$.

C3. There exist a neighborhood $\boldsymbol{V}$ of $\boldsymbol{v_\theta}$ and an integrable function $M(\boldsymbol{s}, \boldsymbol{x})$ with $\mathbb{E}[M(\boldsymbol{s}, \boldsymbol{x})] < \infty$ such that

$$\sup_{\boldsymbol{v} \in \boldsymbol{V}} \|H_{\boldsymbol{\theta}}(\boldsymbol{s}, \boldsymbol{x}; \boldsymbol{v})\|^3 \leq M(\boldsymbol{s}, \boldsymbol{x}), \sup_{\boldsymbol{v} \in \boldsymbol{V}} \left\| \frac{\partial H_{\boldsymbol{\theta}}(\boldsymbol{s}, \boldsymbol{x}; \boldsymbol{v})}{\partial \boldsymbol{v}^\top} \right\| \leq M(\boldsymbol{s}, \boldsymbol{x}).$$

C4. $\mathbb{E}\partial H_{\boldsymbol{\theta}}(\boldsymbol{s}, \boldsymbol{x}; \boldsymbol{v_\theta})/\partial \boldsymbol{v}^\top$ is non-degenerate and $\mathbb{E}[H_{\boldsymbol{\theta}}(\boldsymbol{s}, \boldsymbol{x}; \boldsymbol{v_\theta})H_{\boldsymbol{\theta}}(\boldsymbol{s}, \boldsymbol{x}; \boldsymbol{v_\theta})^\top]$ is positive definite.

The condition C1 ensures $\boldsymbol{v_\theta}$ is identifiable. The conditions C1-C4 guarantee $H_{\boldsymbol{\theta}}(\boldsymbol{s}, \boldsymbol{x}; \boldsymbol{v_\theta})$ is well defined. We define $g_i(\boldsymbol{\sigma}) = (\boldsymbol{s}_i - \bar{\boldsymbol{s}})[d_{\boldsymbol{\theta}}(\boldsymbol{x}_i) - \bar{d}_{\boldsymbol{\theta}}] - \boldsymbol{\sigma}$. The proof of Theorem 3.1 is similar to the argument of Zheng et al. (2012). To prove Theorem 3.1, we need the following three lemmas from Zheng et al. (2012).

**Lemma A.1** *(Lemma A.1. Zheng et al. (2012)) Under the conditions C1–C4, we have that*

$$\max_{1 \leq i \leq n} g_i(\boldsymbol{\sigma_\theta}) = o_p(\sqrt{n}).$$

**Lemma A.2** *(Lemma A.2. Zheng et al. (2012)) Under the conditions C1–C4, we have that $n^{-1}\sum_{i=1}^n g(\boldsymbol{\sigma_\theta})g(\boldsymbol{\sigma_\theta})^\top$ converges in probability to $\Sigma$ as $n \rightarrow \infty$, where $\Sigma$ is the variance-covariance matrix of $g(\boldsymbol{\sigma_\theta})$ .*

**Lemma A.3** *(Lemma A.3. Zheng et al. (2012)) Under the conditions C1–C4, we have that $n^{-1/2}\sum_{i=1}^n g(\boldsymbol{\sigma_\theta})$ converges in distribution to $N(0, \Sigma)$ as $n \rightarrow \infty$.*

By the Lagrange multiplier method, we have

$$p_i = \frac{1}{n} \frac{1}{1 + \boldsymbol{\lambda}^\top g_i(\boldsymbol{\sigma_\theta})},$$

where $\boldsymbol{\lambda}$ satisfies $m$ equations given by

$$h(\boldsymbol{\lambda}) := \frac{1}{n} \sum_{i=1}^n \frac{g_i(\boldsymbol{\sigma_\theta})}{1 + \boldsymbol{\lambda}^\top g_i(\boldsymbol{\sigma_\theta})} = \boldsymbol{0}.$$

Let $\boldsymbol{\lambda} = \rho \boldsymbol{w}$ where $\rho > 0$ and $\|\boldsymbol{w}\| = 1$. Let $g^*(\boldsymbol{\sigma_\theta}) = \max_{1 \leq i \leq n} \|g_i(\boldsymbol{\sigma_\theta})\|$. By Lemma A.1, one has

$$g^*(\boldsymbol{\sigma_\theta}) = o_p(n^{1/2}). \tag{A.1}$$

We have

$$\begin{aligned} 0 &= \|h(\rho \boldsymbol{w})\| \\ &\geq |\boldsymbol{w}^\top h(\rho \boldsymbol{w})| \\ &\geq \frac{\rho \boldsymbol{w}^\top \Sigma_n \boldsymbol{w}}{1 + \rho g^*(\boldsymbol{\sigma_\theta})} - \frac{1}{n} \boldsymbol{w}^\top \sum_{i=1}^n g_i(\boldsymbol{\sigma_\theta}), \end{aligned} \tag{A.2}$$

where

$$\Sigma_n = \frac{1}{n} \sum_{i=1}^{n} g_i(\boldsymbol{\sigma_\theta}) g_i(\boldsymbol{\sigma_\theta})^\top.$$

By Lemma A.3, we have $1/n\boldsymbol{w}^\top \sum_{i=1}^{n} g_i(\boldsymbol{\sigma_\theta}) = O_p(n^{-1/2})$. $\boldsymbol{w}^\top \Sigma_n \boldsymbol{w} \geq \lambda_{min} + o_p(1)$, where $\lambda_{min} > 0$ is the smallest eigenvalue of $\Sigma$. Combining (A.1) and (A.2), it follows that

$$\|\boldsymbol{\lambda}\| = \rho = O_p(n^{-1/2}). \tag{A.3}$$

Let $l_i = \boldsymbol{\lambda}^\top g_i(\boldsymbol{\sigma_\theta})$. We have

$$\max_{i \leq i \leq n} |l_i| = O_p(n^{-1/2}) o_p(n^{1/2}) = o_p(1). \tag{A.4}$$

Combining (6), (A.3) and (A.4), we obtain,

$$\boldsymbol{\lambda} = \Sigma_n^{-1} \left[ \frac{1}{n} \sum_{i=1}^{n} g_i(\boldsymbol{\sigma_\theta}) \right] + o_p(1). \tag{A.5}$$

Therefore,

$$\sum_{i=1}^{n} \boldsymbol{\lambda}^\top g_i(\boldsymbol{\sigma_\theta}) g_i(\boldsymbol{\sigma_\theta})^\top \boldsymbol{\lambda} = \left[ \frac{1}{\sqrt{n}} \sum_{i=1}^{n} g_i(\boldsymbol{\sigma_\theta}) \right]^\top \Sigma_n^{-1} \left[ \frac{1}{\sqrt{n}} \sum_{i=1}^{n} g_i(\boldsymbol{\sigma_\theta}) \right] + o_p(1). \tag{A.6}$$

Taking Taylor expansion at $\boldsymbol{\lambda} = \boldsymbol{0}$, we get from Equations (A.5) and (A.6)

$$\begin{aligned}
-2 \log R(\boldsymbol{\sigma_\theta}) &= 2 \sum_{i=1}^{n} \log\{1 + \boldsymbol{\lambda}^\top g_i(\boldsymbol{\sigma_\theta})\} \\
&= 2 \sum_{i=1}^{n} \boldsymbol{\lambda}^\top g_i(\boldsymbol{\sigma_\theta}) - \sum_{i=1}^{n} \boldsymbol{\lambda}^\top g_i(\boldsymbol{\sigma_\theta}) g_i(\boldsymbol{\sigma_\theta})^\top \boldsymbol{\lambda} + o_p(1) \\
&= \left[ \frac{1}{\sqrt{n}} \sum_{i=1}^{n} g_i(\boldsymbol{\sigma_\theta}) \right]^\top \Sigma_n^{-1} \left[ \frac{1}{\sqrt{n}} \sum_{i=1}^{n} g_i(\boldsymbol{\sigma_\theta}) \right] + o_p(1).
\end{aligned} \tag{A.7}$$

By condition C3 and Lemma A.1, the last term in (A.7) is $o_p(1)$. Thus, we have

$$-2 \log R(\boldsymbol{\sigma_\theta}) \xrightarrow{D} \chi_m^2$$

by using Lemmas A.2 and A.3.

## B EXAMPLE

As an example, we present the use of our approach with the binary logistic regression.

**Example B.1** *Binary Logistic Regression. In a binary logistic regression, the objective is to predict the binary label $y$ by the features $\boldsymbol{x}$. The classifier decision boundary is defined as $d_{\boldsymbol{\theta}}(\boldsymbol{x}) = \boldsymbol{\theta}^\top \boldsymbol{x}$. The sensitive vector is denoted by $\boldsymbol{s}$, and the goal is to ensure fairness with respect to this sensitive attribute vector. The logistic regression model assumes a probability distribution as follows,*

$$p(y = 1 | \boldsymbol{x}, \boldsymbol{\theta}) = \frac{1}{1 + e^{-\boldsymbol{\theta}^\top \boldsymbol{x}}}, \; p(y = -1 | \boldsymbol{x}, \boldsymbol{\theta}) = \frac{e^{-\boldsymbol{\theta}^\top \boldsymbol{x}}}{1 + e^{-\boldsymbol{\theta}^\top \boldsymbol{x}}}.$$

*The loss function is defined as $L(\boldsymbol{\theta}) = -\sum_{i=1}^{n} \log p(y_i | \boldsymbol{x}_i, \boldsymbol{\theta})$. Therefore, the fair logistic regression by EL can be obtained by solving the following problem,*

$$\begin{aligned}
\min_{\boldsymbol{\theta} \in \boldsymbol{\Theta}, \boldsymbol{\lambda} \in \mathbb{R}^m} &- \sum_{i=1}^{n} \log p(y_i | \boldsymbol{x}_i, \boldsymbol{\theta}) \\
s.t. \; &2 \sum_{i=1}^{n} \log\{1 + \boldsymbol{\lambda}^\top (\boldsymbol{s}_i - \bar{\boldsymbol{s}})[\boldsymbol{\theta}^\top \boldsymbol{x}_i - \bar{\boldsymbol{x}}]\} \leq \chi_m^2(\alpha), \\
&\frac{1}{n} \sum_{i=1}^{n} \frac{(\boldsymbol{s}_i - \bar{\boldsymbol{s}})(\boldsymbol{\theta}^\top \boldsymbol{x}_i - \boldsymbol{\theta}^\top \bar{\boldsymbol{x}})}{1 + \boldsymbol{\lambda}^\top (\boldsymbol{s}_i - \bar{\boldsymbol{s}})(\boldsymbol{\theta}^\top \boldsymbol{x}_i - \boldsymbol{\theta}^\top \bar{\boldsymbol{x}})} = \boldsymbol{0}.
\end{aligned}$$

## C    COVERAGE PROBABILITY AND CONFIDENCE INTERVAL

Table 5 shows the coverage probability and confidence interval for the scenarios, $\boldsymbol{\theta} = (2,1)^\top$ with $\boldsymbol{\sigma_\theta} = 2$, and $\boldsymbol{\theta} = (3,2)^\top$ with $\boldsymbol{\sigma_\theta} = 7/2$ .

Table 5: Coverage probability and confidence interval

| $n$ | $\boldsymbol{\theta} = (2,1)^\top$ | | | | $\boldsymbol{\theta} = (3,2)^\top$ | | | |
|------|-------|-------|-------|-------|-------|-------|-------|-------|
| | CP | LB | UB | AL | CP | LB | UB | AL |
| 100 | 0.933 | 1.352 | 2.813 | 1.461 | 0.936 | 2.434 | 4.846 | 2.413 |
| 200 | 0.943 | 1.520 | 2.557 | 1.037 | 0.946 | 2.708 | 4.421 | 1.713 |
| 500 | 0.939 | 1.684 | 2.341 | 0.657 | 0.939 | 2.978 | 4.064 | 1.085 |
| 800 | 0.943 | 1.747 | 2.266 | 0.520 | 0.945 | 3.082 | 3.941 | 0.859 |
| 1200 | 0.945 | 1.791 | 2.215 | 0.424 | 0.943 | 3.155 | 3.856 | 0.701 |

