# OpenReview forum: "Empirical Likelihood for Fair Classification"
_ICLR.cc/2024/Conference — ICLR 2024 poster_

### Official Review · Reviewer_umPv · 2023-10-18

**Soundness:** 3 good
**Presentation:** 2 fair
**Contribution:** 2 fair
**Rating:** 5
**Confidence:** 3

**Summary:**

This paper proposes to use the covariance as a proxy for the fairness and develop the confidence region of the covariance vector using empirical likelihood. In this way, a confidence region (with the significant level alpha being used for the fairness constraint) can be used to provide a more interpretable way to trade-off between accuracy and fairness (DP).

**Strengths:**

1. The proposed approach is technically sound and does seem to have the potential to replace the covariance based measure in Zafar et al., 2017 and become a more general measurement for DP.

2. Comparing results from Table 3 and 4, the proposed algorithm does have a positive impact on both DP and EO on the ACS PUMS datasets including the OOD cases.

**Weaknesses:**

The proposed method itself is technically sound and has good potential. Most of my concerns are regarding the evaluation of the method in the experiments.

1. The simulation results in Table 2 do not seem to be a fair comparison. I would suggest adding more granularity for Zafar et al. with 0<c<0.1.

2. In Table 3, increasing alpha from 0.05 to 0.5 does not seem to affect the performance much (both in training and testing).

3. The proposed algorithm should be tested on more commonly used fairness datasets such as [1] or the ones from UCI Machine Learning Repository (https://archive.ics.uci.edu/)--- Adult Census Income, German Credit, etc.

4. Only one baseline is compared in the experiments. I would suggest the authors to test against other baseline approaches optimizing for DP.

[1] Ding, Frances, Moritz Hardt, John Miller, and Ludwig Schmidt. "Retiring adult: New datasets for fair machine learning." Advances in neural information processing systems 34 (2021): 6478-6490.

**Questions:**

1. Can you provide more simulation results in Table 2 for Zafar et al. with 0<c<0.1? E.g. c=0.01, 0.02, ... Meanwhile, can alpha take values of 1? Or what will happen when alpha is close to 1, e.g. alpha = 0.99?

2. In Table 3, increasing alpha from 0.05 to 0.5 does not seem to affect the performance much (both in training and testing). Is it still the case when alpha is increased to e.g. 0.9?

---

> ### Author Response · Authors · 2023-11-18
>
> Thank you very much for your helpful comments.
>
> Response to Weakness 1：We have included results for Zafar et al. (2017) with $0<c<0.1$ in our response to Question 1.
>
> Response to Weakness 2: The impact of increasing $\alpha$ has been discussed in our response to Question 2.
>
> Response to Weakness 3: We have applied our method to these datasets. In Section 5.1 of our paper, we use the data (Ding et al., 2021) to show the robust of our method to distribution shifts. In Section 5.2, we use the German credit dataset to test our method by considering two sensitive features simultaneously. We kindly request the reviewer to revisit these sections for additional insights.
>
>     Ding, F., Hardt, M., Miller, J., & Schmidt, L. (2021). Retiring adult: New datasets for fair machine learning. In Advances in Neural Information Processing Systems. Curran Associates, Inc.
>
> Response to Weakness 4：Our experiments have been expanded by comparing with a method that directly optimizes DP (Weerts et al., 2023). Please refer to our global response for the new experimental results.
>
>     Weerts, H.; Dud´ık, M.; Edgar, R.; Jalali, A.; Lutz, R.; Madaio, M. Fairlearn: Assessing and improving fairness of ai systems. Journal of Machine Learning Research, 24(257):1–8, 2023.
>
> Response to Question 1:  In the following table, we present the outcomes of Zafar et al. (2017) with $0<c<0.1$. Notably, none of these results exhibit superior performance in both accuracy and fairness metrics when compared with our method. Our method achieves a higher (or equal) accuracy and a higher fairness than Zafar et al. (2017): $\alpha=0.5$ vs. $c=0.035$, $\alpha=0.1$ vs. $c=0.095$, $\alpha=0.2$ vs. $c=0.095$.
>
> | c     | ACC  | F1    | p%      | DP    | EO    |
> |-------|-------|-------|--------|--------|-------|
> | 0.015 | 0.832 | 0.833 | 89.941 | -0.016 | 0.109 |
> | 0.035 | 0.835 | 0.836 | 86.465 | 0.038  | 0.149 |
> | 0.055 | 0.842 | 0.842 | 83.077 | 0.050  | 0.188 |
> | 0.075 | 0.843 | 0.843 | 81.346 | 0.065  | 0.207 |
> | 0.095 | 0.842 | 0.840 | 78.548 | 0.094  | 0.240 |
>
> As $\alpha$ is closer to 1, the constraint becomes more strict. Theoretically, $\alpha$ can take the value of 1. When $\alpha=1$, the length of the confidence interval becomes 0, which means the confidence interval only includes one point 0. We show the result when $\alpha=0.99$ in the following table.
>
>  | $\alpha$     | ACC  | F1    | p%      | DP    | EO    |
> |-------|-------|-------|--------|--------|-------|
> | 0.99 | 0.825 | 0.827 | 92.308 | -0.029 | 0.083 |
>
>  Response to Question 2: The initial glance may suggest only marginal fairness enhancement when increasing $\alpha$ from 0.05 to 0.5. However, when assessing the balance between accuracy loss and fairness improvement, we have a more detailed perspective. We calculate the average AC, DP, and EO reductions across the eight states. On average, AC decreases 0.0022, DP decreases 0.0150, and EO decreases 0.0185 for the training data. The ratios of fairness improvement to accuracy loss are more significant, with values of 0.0150/0.0022=6.78 for DP and 0.0185/0.0022=8.34 for EO. For the test data, AC decreases 0.0019, DP decreases 0.0156, and EO decreases 0.0132 on average, with ratios 0.0156/0.0019=8.28 for DP, 0.0132/0.0019=7.01 for EO.
>
> In addition, we increase $\alpha$ to 0.9 as shown in the following table. We compare the results between $\alpha=0.05$ and $\alpha=0.9$. On average, AC decreases 0.0028, DP decreases 0.0207, and EO decreases 0.0224 for the training data. The ratios are 0.0207/0.0028=7.28 for DP and 0.0224/0.0028=7.86 for EO. For the test data, AC decreases 0.0027, DP decreases 0.0214, and EO decreases 0.0194 on average, with ratios 0.0214/0.0027=7.79 for DP, 0.0194/0.0027=7.07 for EO. These findings underscore the effective role of $\alpha$ in balancing the accuracy and fairness, where fairness gains are substantial comparing to the minor accuracy loss.
>
>  |         |  |  $\alpha$ |   =     |     0.9   |            ||
> |-------|-------------|-------------------|--------|--------|------------|------------|
> |    |       ID       | (training)     |        | OOD     |   (test)  |        |
> | State | ACC          | DP                | EO     | ACC     | DP         | EO     |
> | CA                | 0.7239      | 0.1967            | 0.2259 | 0.7282 | 0.2020     | 0.2111 |
> | TX                | 0.7330      | 0.2195            | 0.2605 | 0.7337 | 0.1874     | 0.1956 |
> | DE                | 0.7290      | 0.2186            | 0.2882 | 0.7375 | 0.2595     | 0.2538 |
> | NV                | 0.7152      | 0.1874            | 0.1948 | 0.7312 | 0.1611     | 0.1518 |
> | KY                | 0.7451      | 0.1413            | 0.1464 | 0.7181 | 0.0759     | 0.1225 |
> | FL                | 0.7376      | 0.2083            | 0.2402 | 0.7295 | 0.1546     | 0.1669 |
> | MO                | 0.7455      | 0.1399            | 0.2025 | 0.7220 | 0.0917     | 0.1319 |
> | NE                | 0.7297      | -0.0411           | 0.0317 | 0.7109 | 0.0354     | 0.0972 |

---

### Official Review · Reviewer_Rn9g · 2023-10-30

**Soundness:** 3 good
**Presentation:** 3 good
**Contribution:** 3 good
**Rating:** 6
**Confidence:** 3

**Summary:**

Based on the covariance fairness measure by Zafar et al. (2017), the authors propose a fairness constraint based on a statistical likelihood test. Specifically, they formulate a constraint that requires that a statistical test with level alpha does not reject $H_0$: the classifier is fair.

**Strengths:**

I think that the idea of using a likelihood statistic for enforcing fairness is novel and an interesting approach. The paper is well written and supported by the required theoretical developments.

**Weaknesses:**

I think the main limitation of this work is the experimental evaluation. I understand that the method is based on Zafar's covariance fairness measure, and comparing to that is very valuable. However, since this seminal work in 2017, there has been a lot of development in the community regarding fairness regularizers or constraints. Comparing the results of your approach to some of these more recent works would be of interest.

**Questions:**

Some minor questions and comments:
- On the bottom of page 3, you state that a system of $m$ constraints and $d$ parameters with $m\geq d$ can only have the solution $\theta=0$. If I understand correctly $m$ is the number of sensitive attributes and $d$ is the number of features. Does it make sense that $m\geq d$?
- In equation (8), is $\lambda$ a decision variable of the minimization problem?
- Am I correct in my analysis that (8) is nonconvex?
- I think Figure 1 is very hard to understand. Should we compare Figure 1 with Figure 2 and observe that the ID results are more representative of the OOD resutls than in Figure 2?

---

> ### Author Response · Authors · 2023-11-18
>
> Thank you very much for your helpful comments.
>
> Response to Weakness: We have significantly expanded our experiments by comparing with the method (Weerts et al. 2023) in our global response. The results showcase our EL method is superior to the compared approach.  We kindly request the reviewer's attention to the new experimental results provided in our global response.
>
>     Weerts, H.; Dud´ık, M.; Edgar, R.; Jalali, A.; Lutz, R.; Madaio, M. Fairlearn: Assessing and improving fairness of ai systems. Journal of Machine Learning Research, 24(257):1–8, 2023.
>
> Response to Question 1: $m$ is the number of sensitive attributes and $d$ is the number of non-sensitive features. There are some cases where $m\geq d$.  To illustrate this situation, consider the Ricci 20 dataset (Le et al., 2022). This dataset encompasses five features "race", "position", "written", "oral", and the derived feature "combine" representing a composite score based on "written" and "oral", and one label "promoted". If we only consider the four original features and study the discrimination of race and position on the promotion, we have two sensitive features "race" and "position" and two non-sensitive features "written" and "oral". Then $m=d=2$.
>
>     Le Quy, T., Roy, A., Iosifidis, V., Zhang, W., & Ntoutsi, E. (2022). A survey on datasets for fairness-aware machine learning. WIREs Data Mining and Knowledge Discovery, 12(3), e1452.
>
>  Response to Question 2: Yes. In equation (8), $\lambda$ is a decision variable of the minimization problem.
>
>  Response to Question 3: The optimization problem in equation (8) is nonconvex. We can obtain a local minimum when dealing with nonconvex problems. Our empirical experiments confirm that this approach yields satisfactory performance. Achieving a global minimum in nonconvex problems is indeed a demanding task, and it falls outside the scope of our focus in this paper.
>
>  Response to Question 4: The in-distribution (ID) results represent the training outcomes, and the out-of-distribution (OOD) results indicate the performance across 49 other states during the testing. Figure 2 (adapted from Figure 3 of Ding et al. (2021)) shows the accuracy of the training data (ID) is much higher than the accuracy among all the test data (OOD), which underscores a marked separation between the in-distribution and out-of-distribution results. In contrast, our results in Figure 1, depict OOD outcomes distributed around the ID results in a random pattern without a marked separation between ID and OOD. The key point in comparison between Figure 1 and Figure 2 is that Figure 2 demonstrates a marked separation between in-distribution (ID) and out-of-distribution (OOD) results, while Figure 1 not. We can say that the ID results are more representative of the OOD results in Figure 1 than in Figure 2.

---

> > ### Comment · Reviewer_Rn9g · 2023-11-20
> > **Thank you for your comments**
> >
> > Thank you for your clarifications. The new experiments are insightful and show that the proposed method performs well.
> > Some follow-up questions on your answers to my questions:
> > Q1: So you assume the protected attribute is not part of the feature vector?
> > Q2: Thank you for the clarification. I suggest adding $\lambda$ under the $\min$, like $\theta$, to avoid confusion.
> > Q3: That makes sense. Is there a straight-forward way to apply your methodology to (S)GD type algorithms (i.e. Neural Networks)
> > Q4: Thank you, that is clear now.

---

> ### Author Response · Authors · 2023-11-20
> **Response to new comments**
>
> Thanks for your helpful feedback.
>
> Response to Q1: Yes. The protected attribute is excluded from the feature vector in our methodology, aligning with the approach employed by Zafar et al. (2017) and Zink and Rose (2020).
>
>     Zafar, M.B., Valera, I., Rogriguez, M.G. &amp; Gummadi, K.P.. (2017). Fairness Constraints: Mechanisms for Fair Classification. Proceedings of the 20th International Conference on Artificial Intelligence and Statistics.
>
>     Zink A, Rose S. 2020 . Fair regression for health care spending. Biometrics. 76(3):973-982.
>
> Response to Q2: Thanks for your suggestion. We have added $\lambda$ under "min" in our revised paper.
>
> Response to Q3:  Yes. our methodology can be applied to ((S)GD) type algorithms, including Neural Networks. To implement this, one can leverage packages like tensorflow_constrained_optimization (tfco), specifically designed for integrating gradient-based learning algorithms such as SGD or Adam for solving constrained optimization problems. The tfco package provides flexibility in choosing different gradient-based learning algorithms, making it compatible with our proposed methodology.
>
> Response to Q4: We are glad the explanation is clear now. If you have any further questions or need additional clarification, feel free to ask.

---

> > ### Comment · Reviewer_Rn9g · 2023-11-20
> >
> > Thank you for clarifying. I would like to retain my score.

---

### Official Review · Reviewer_YzKS · 2023-11-04

**Soundness:** 3 good
**Presentation:** 3 good
**Contribution:** 3 good
**Rating:** 6
**Confidence:** 4

**Summary:**

This paper proposes a framework to ensure fairness of classifiers while handling the uncertainty from point estimates using finite samples. In particular, formulating fairness in terms of the covariance between sensitive attributes and the decision boundary (Zafar et al. 2017), the authors propose using the empirical likelihood method to derive a confidence region for the covariance vector. Empirical evaluation using simulation as well as real-world data shows that the proposed approach can balance the accuracy-fairness tradeoff and be robust to distribution shifts.

**Strengths:**

The consideration of uncertainty in fairness assessment is interesting, and addresses a problem that is underexplored in existing approaches.

The application of an EL-based estimator for fairness using confidence intervals is a novel contribution. The authors suggest that this approach could be applied to other fairness criteria, which could increase the impact of this method broadly.

The paper is overall well-written and technically sound as far as I can tell.

**Weaknesses:**

While the comparison to Zafar et al. (2017) is very thorough, no other fair learning methods are considered in the empirical evaluation. In particular, comparing group fairness metrics such as DP and EO against baselines that directly optimize those notions could serve to more strongly justify the proposed approach using covariance-based fairness constraints.

I was a bit confused about the discussion at the end about applying EL to other fairness criteria. If indeed extending to non-linear measures is straightforward, it is unclear why the current method only considers covariance as a fairness proxy.

The motivation for the covariance-based fairness constraints was not very compelling. The authors suggest that determining the threshold for accuracy-fairness tradeoff becomes easier with the proposed approach. However, it is still unclear what the appropriate confidence region is for a given group fairness notion (whether a soft constraint to bound the violation or as a hard constraint).

**Questions:**

1. How does the proposed method compare against Zafar et al. (2017) under distribution shifts?

2. Is the discussion about EL for general fairness measures new or known results? If it is the former, there needs to be more details and proofs.

3. I was not sure about the significance of Figure 1. Also, should the legends ID and OOD evaluations be switched in Figures 1 and 2? Figure 2 was also extremely hard to read.

---

> ### Author Response · Authors · 2023-11-18
>
> Thank you very much for your helpful comments.
>
> Response to Weakness 1：Our experiments have been expanded by comparing with a method that directly optimizes DP (Weerts et al., 2023). Please refer to our global response for the new experimental results.
>
>     Weerts, H.; Dud´ık, M.; Edgar, R.; Jalali, A.; Lutz, R.; Madaio, M. Fairlearn: Assessing and improving fairness of ai systems. Journal of Machine Learning Research, 24(257):1–8, 2023.
>
> Response to Weakness 2：We can extend EL to other fairness criteria using similar techniques as presented in our paper. However, the extension involves carefully finding the influence functions and deriving the estimating equations of the specific fairness criteria, which serves as the foundation of the logarithm of the EL ratio converging to the $\chi^2$ distribution. Our current paper represents a preliminary exploration of applying EL to the fairness metrics. We leave the extension of EL to other fairness criteria as a future project.
>
> Response to Weakness 3：The tradeoff indicator $\alpha$ in our framework has a bounded range from 0 to 1, while covariance is unbounded. In this sense, it is easier to determine the threshold by our method. The choice of the confidence region depends the preference of the user's accuracy and fairness. Our constraint is formed by imposing $0$ included in the confidence region of the covariance at a user-determined significant level. The inherent mechanism of the confidence region allows the violation of the covariance as $0$, controlled by $\alpha$. In this sense, our constraint can be seen as a soft constraint.
>
> Response to Question 1: Our method performs better than the method in Zafar et al. (2017) under distribution shifts. We conducted experiments using datasets characterized by explicit distribution shifts (Ding et al., 2021), and the results are shown below. In each row of the following table, our method performs better in both accuracy and fairness than Zafar et al. (2017).
>
> |       |  Zafar|  et al.| (2017) |        | EL    | based  |fairness|        |
> |-------|-------|--------|--------|--------|-------|--------|--------|--------|
> | State | $c$   | ACC    | DP     | EO     | $\alpha$ | ACC | DP     | EO     |
> | CA    | 0.007 | 0.7284 | 0.2036 | 0.2125 | 0.50  | 0.7284 | 0.2034 | 0.2123 |
> | TX    | 0.020 | 0.7348 | 0.2009 | 0.2110 | 0.01  | 0.7348 | 0.1983 | 0.2106 |
> | TX    | 0.006 | 0.7340 | 0.1909 | 0.1993 | 0.50  | 0.7340 | 0.1905 | 0.1988 |
> | TX    | 0.003 | 0.7338 | 0.1893 | 0.1978 | 0.70  | 0.7338 | 0.1892 | 0.1975 |
> | NV    | 0.020 | 0.7319 | 0.1686 | 0.1598 | 0.50  | 0.7319 | 0.1657 | 0.1542 |
> | NV    | 0.015 | 0.7317 | 0.1665 | 0.1595 | 0.50  | 0.7319 | 0.1657 | 0.1542 |
> | NV    | 0.009 | 0.7314 | 0.1642 | 0.1549 | 0.70  | 0.7314 | 0.1634 | 0.1535 |
> | FL    | 0.008 | 0.7304 | 0.1605 | 0.1774 | 0.30  | 0.7305 | 0.1605 | 0.1771 |
>
> The following table shows the results of Zafar et al. (2017) whe $c=0.01$ under distribution shifts. The fairness metrics differ significantly between the training data and test data, which indicates the method is not robust to distribution shifts.
>
> |       |   ID        |results  |(training)|OOD    |results |(test)  |
> |-------|-------------|---------|---------|--------|--------|--------|
> | State | ACC         | DP      | EO      | ACC    | DP     | EO     |
> | CA    | 0.7247      | -0.2286 | -0.1996 | 0.7286 | 0.2053 | 0.2130 |
> | TX    | 0.7341      | -0.2640 | -0.2193 | 0.7343 | 0.1936 | 0.2038 |
> | DE    | 0.7288      | -0.2882 | -0.2191 | 0.7378 | 0.2620 | 0.2620 |
> | NV    | 0.7161      | -0.1969 | -0.1893 | 0.7314 | 0.1645 | 0.1564 |
> | KY    | 0.7458      | -0.1495 | -0.1443 | 0.7197 | 0.0823 | 0.1273 |
> | FL    | 0.7396      | -0.2451 | -0.2123 | 0.7307 | 0.1619 | 0.1795 |
> | MO    | 0.7459      | -0.2043 | -0.1434 | 0.7239 | 0.1084 | 0.1427 |
> | NE    | 0.7306      | -0.0336 | 0.0182  | 0.7123 | 0.0426 | 0.1056 |
>
>     Ding, F., Hardt, M., Miller, J., & Schmidt, L. (2021). Retiring adult: New datasets for fair machine learning. In Advances in Neural Information Processing Systems. Curran Associates, Inc.
>
> Response to Question 2: The extension of EL for general fairness measures can be considered as new results, as explained in our response to Weakness 2. We leave the extension as a future project.
>
> Response to Question 3:  The legends ID and OOD evaluations should be switched in Figure 1. We apologize for our carless errors. We have corrected the errors in our revised paper. The in-distribution (ID) results represent the training outcomes, and the out-of-distribution (OOD) results indicate the performance across 49 other states during testing. Figure 2 (Ding et al., 2021) shows the accuracy of the training data is much higher than the accuracy among all the test data, which underscores a marked separation between the ID and OOD results. In contrast, our results in Figure 1 depict OOD outcomes distributed around ID results in a random pattern without a marked separation between ID and OOD.

---

### Author Response · Authors · 2023-11-18
**Global response**

We extend our sincerest appreciation to the reviewers for their invaluable insights and constructive comments. In response to these valuable suggestions, we conducted additional experiments to rigorously evaluate the effectiveness of our proposed EL method. Specifically, we augmented our experiments by comparing our approach with a method directly optimizing for Demographic Parity (DP) (Weerts et al., 2023). We employed multiple datasets, including the Adult dataset (Aldut), the Bank Marketing dataset (Bank), the Boston Housing dataset (Housing), the 'Default of Credit Card Clients' dataset (Credit), and the Diabetes 130-Hospitals dataset (Diabetes). Since the compared method is specifically optimized for DP, we adjust the hyperparameter $\alpha$ in our method to optimize for a better DP. The summarized results are presented in the table below, where, for each dataset, our method (EL-based fairness) consistently achieves higher accuracy (ACC) and superior fairness (DP) compared to the method directly optimizing for DP (Optimized DP). This robust performance across diverse datasets underscores the superiority of our proposed approach.

|   Dataset     |    Optimized|DP|EL-based|fairness|
|----------|--------|--------|--------|--------|
|          | ACC    | DP     | ACC    | DP     |
| Aldut    | 0.7660 | 0.0219 | 0.7920 | 0.0191 |
| Bank     | 0.7240 | 0.0482 | 0.7260 | 0.0442 |
| Housing  | 0.8542 | 0.2188 | 0.8750 | 0.2031 |
| Credit   | 0.8080 | 0.0254 | 0.8200 | 0.0206 |
| Diabetes | 0.7200 | 0.0273 | 0.7560 | 0.0119 |

    Weerts, H.; Dud´ık, M.; Edgar, R.; Jalali, A.; Lutz, R.; Madaio, M. Fairlearn: Assessing and improving fairness of ai systems. Journal of Machine Learning Research, 24(257):1–8, 2023.
    R. Kohavi and B. Becker, UCI Machine Learning Repository: Adult Data Set, 01-May-1996.
    S. Moro, P. Cortez, and P. Rita, UCI Machine Learning Repository: Bank Marketing Data Set, 14-Feb-2014.
    D. Harrison and D. L. Rubinfeld, “Hedonic housing prices and the demand for clean air,” Journal of Environmental Economics and Management, vol. 5, no. 1, pp. 81–102, Mar. 1978.
    I-Cheng Yeh and Che-hui Lien, “The comparisons of data mining techniques for the predictive accuracy of probability of default of credit card clients”, Expert Systems with Applications, 36(2), 2473-2480, 2009.
    Beata Strack, Jonathan Deshazo, Chris Gennings, Juan Luis Olmo Ortiz, Sebastian Ventura, Krzysztof Cios, and John Clore. Diabetes 130-us hospitals for years 1999-2008 data set. 05 2014.

---

### Meta-Review · Area_Chair_s5Z1 · 2023-12-08

**Metareview:**

a) claims: This paper extends an existing covariance based fairness framework by estimating a confidence region for the covariance matrix.  The learned classifier is considered fair if the null hypothesis of 0 covariance between sensitive attributes and decisions fails to be rejected at a user-specific confidence level.  The fairness of the resulting classifiers is evaluated empirically on a number of datasets.

b) strengths: There was a broad consensus that the idea of incorporating uncertainty into fairness evaluations was a valuable contribution, that the theoretical content was strong, and that the paper was clearly written.

c) weaknesses: There was widespread concern that the experimental evaluations were not sufficiently extensive.  In the discussion period, the authors presented additional experimental evaluations that addressed many of these concerns to the reviewers' satisfaction.

**Justification For Why Not Higher Score:**

Reviewers all had a single concern (experimental evaluations)

**Justification For Why Not Lower Score:**

Reviewers were broadly satisfied with the paper

---

### Decision · Program_Chairs · 2024-01-16

Accept (poster)